# DeepRAG: Thinking to Retrieve Step by Step for Large Language Models

**Xinyan Guan[1,2], Jiali Zeng[3,†], Fandong Meng[3], Chunlei Xin[1,2], Yaojie Lu[1],**
**Hongyu Lin[1,†], Xianpei Han[1], Le Sun [1], Jie Zhou[3]**
[1]Chinese Information Processing Laboratory, Institute of Software, Chinese Academy of Sciences
[2]University of Chinese Academy of Sciences
[3]Pattern Recognition Center, WeChat AI, Tencent Inc, China
{guanxinyan2022,chunlei2021,hongyu,luyaojie,xianpei,sunle}@iscas.ac.cn
{lemonzeng,fandongmeng,withtomzhou}@tencent.com

## Abstract

Large Language Models (LLMs) have shown remarkable reasoning capabilities, while their practical applications are limited by severe factual hallucinations due to limitations in the timeliness, accuracy, and comprehensiveness of their parametric knowledge. Meanwhile, enhancing retrieval-augmented generation (RAG) with reasoning remains challenging due to ineffective task decomposition and redundant retrieval, which can introduce noise and degrade response quality. In this paper, we propose DeepRAG, a framework that models retrieval-augmented reasoning as a Markov Decision Process (MDP), enabling reasonable and adaptive retrieval. By iteratively decomposing queries, DeepRAG dynamically determines whether to retrieve external knowledge or rely on parametric reasoning at each step. Experiments show that DeepRAG improves retrieval efficiency and boosts answer accuracy by 25.41%, demonstrating its effectiveness in enhancing retrieval-augmented reasoning. [1]

## 1 Introduction

Large Language Models (LLMs) have shown considerable promise in reasoning (Plaat et al., 2024). Nevertheless, their limitations in capacity and capabilities result in significant issues with factual hallucinations, stemming from challenges related to the timeliness, accuracy, and comprehensiveness of their parametric knowledge (Zhang et al., 2023; Huang et al., 2023). To mitigate these problems, Retrieval-Augmented Generation (RAG) has been introduced as a promising approach. By incorporating relevant information from knowledge bases or search engines, RAG enhances the factual accuracy of model responses (Zhao et al., 2024).

However, enhancing RAG with reasoning still poses several challenges (Gao et al., 2025). One significant issue is that complex queries often necessitate multi-step decomposition to establish a coherent reasoning process (Radhakrishnan et al., 2023; Guan et al., 2024). Iterative retrieval has been proposed as a solution to continuously update retrieval results, addressing the dynamic information needs that arise during the generation process (Yue et al., 2024; Wang et al., 2025). Despite this, LLMs frequently struggle to generate precise and atomic subqueries, which are essential for more effective retrieval and question decomposition (Wu et al., 2024). From the perspective of RAG, iterative retrieval should ideally generate the next atomic query based on the current question and the available information in an adaptive manner. As illustrated in Figure 1, the process flows logically from one step to the next. Specifically, the goal of finding each movie's runtime in *steps 2-4* is derived from *step 1*'s identification of the three titles of the Lord of the Rings series.

Additionally, retrieval is not always essential (Jeong et al., 2024). Some queries depend on external knowledge (*steps 2-4*), while others can be addressed through the reasoning capabilities of the LLM alone (*step 5* requires summarizing previous information). Moreover, LLMs have shown the ability

---

[†]Corresponding author.
[1]https://github.com/icip-cas/DeepRAG

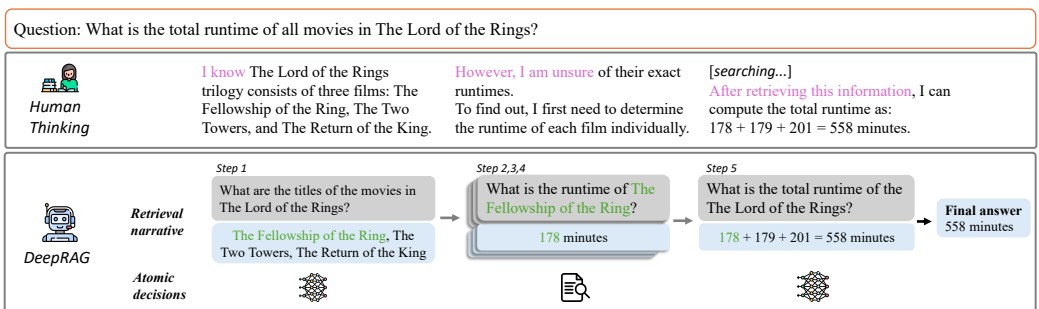

Figure 1: Correspondence between human thinking processes and DeepRAG. Specifically, *retrieval narrative* ensures a structured workflow by generating subqueries that seek additional information based on previous content, and *atomic decisions* dynamically determines whether to retrieve external knowledge or rely solely on the parametric knowledge for each subquery.

to function as knowledge bases in their own right (Petroni et al., 2019) (such as in *step 1*, where the three movie titles are widely known). Unnecessary retrieval can be redundant and may introduce noise, and degrade the quality of generated responses (Chen et al., 2023; Tan et al., 2024; Yu et al., 2022).

To tackle these issues, we introduce **DeepRAG**, a new framework inspired by how humans search the Internet based on demand. This framework aims to enhance reasoning capabilities in retrieval-augmented generation by modeling the process as a Markov Decision Process. DeepRAG incorporates two main components: *retrieval narrative* and *atomic decisions*, which together create a strategic and adaptive retrieval system. As depicted in Figure 1, the *retrieval narrative* ensures a structured workflow by generating subqueries that seek additional information based on previous content. For each subquery, *atomic decisions* dynamically determines whether to retrieve external knowledge or rely solely on the LLM's parametric knowledge.

As illustrated in Figure 2, our framework consists of three components: 1) **Binary Tree Search**, which constructs a binary tree for each subquery related to the question, exploring paths based on parametric knowledge or external knowledge. 2) **Imitation Learning**, which extracts the reasoning process that leads to the correct final answer with minimal retrieval cost based on Binary Tree Search, enabling the model to learn the pattern of "subquery generation – *atomic decision* – intermediate answer". 3) **Chain of Calibration**, which enhances the LLM's ability to calibrate its internal knowledge, allowing more accurate *atomic decisions* on when retrieval is necessary. Specifically, we implement two calibration variants tailored for different learning paradigms. For offline calibration, we leverage binary tree search to construct process-supervised signals that guide step-wise optimization. For online calibration, we employ outcome-based rewards to enable autonomous self-exploration and improvement. By explicitly enhancing the LLM's ability to recognize its own knowledge limits, we can train any model in an end-to-end manner, allowing it to dynamically decide when and what to retrieve.

We validate the effectiveness of DeepRAG across in-distribution, out-of-distribution, and heterogeneous knowledge base datasets. Experimental results show that DeepRAG significantly outperforms existing methods, achieving a 25.41% increase in accuracy while also enhancing retrieval efficiency. Further analysis indicates that DeepRAG demonstrates a stronger correlation between its retrieval decisions and parametric knowledge, suggesting more effective calibration of knowledge boundaries.

## 2 RELATED WORK

**Adaptive Retrieval-Augmented Generation** Existing adaptive RAG approaches can be broadly categorized into three types: classifier-based methods (Cheng et al., 2024; Jeong et al., 2024) requiring additional linear head training for retrieval decisions, confidence-based methods (Jiang et al., 2023; Su et al., 2024; Dhole, 2025) relying heavily on threshold-dependent uncertainty metrics, and LLM-based methods (Asai et al., 2023; Zhang et al., 2024) generating retrieval decisions but often fail to accurately recognize their knowledge boundaries, making it unreliable to delegate retrieval timing decisions to the model. Our method leverages the inherent generative capabilities of LLMs to explore

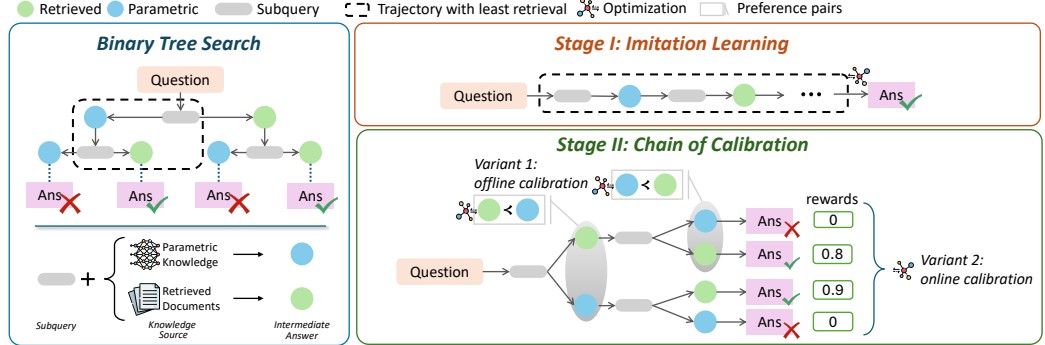

Figure 2: An overview of DeepRAG, our framework comprises three steps: (1) Binary Tree Search, (2) Imitation Learning, and (3) Chain of Calibration. Given a set of supervised datasets, we first use binary tree search to synthesize data for imitation learning, allowing the model to learn effective retrieval patterns. Next, we can either apply offline calibration or online calibration to further calibrate the LLM's awareness of its knowledge boundaries.

knowledge boundaries in RAG settings. This design maintains the model's native generation abilities while eliminating the need for additional parameters or unreliable uncertainty metrics.

**Reasoning in Retrieval-Augmented Generation**  Recent advances in RAG have increasingly emphasized the integration of reasoning capabilities. Search-o1 (Li et al., 2025) incorporates retrieval into inference to build an agentic system, while its application is limited to reasoning models (Chen et al., 2025). Self-RAG (Asai et al., 2023) and Auto-RAG (Yu et al., 2024) enhance reasoning through automatic data synthesis within retrieval-augmented frameworks, while AirRAG (Feng et al., 2025) combines Monte Carlo Tree Search with self-consistency techniques. These methods, however, often depend heavily on extensive retrieval or sampling overhead. More recent developments have explored reinforcement learning to enhance retrieval quality (Jin et al., 2025; Song et al., 2025a; Gao et al., 2024), while these methods generally overlook retrieval efficiency in their reward function. In contrast, DeepRAG offers a flexible, end-to-end solution that enables arbitrary models to retrieve information step by step as needed, based on their evolving reasoning process.

**Knowledge Boundary**  LLMs struggle to accurately distinguish between what they know and what they don't know (Yin et al., 2023; Kapoor et al., 2024a; Yin et al., 2024). Additional fine-tuning (Kapoor et al., 2024b), precise probing (Cheng et al., 2024), or activation control (Xin et al., 2025) is typically required to calibrate the model's cognition. Our approach explores knowledge boundaries in RAG settings.

## 3 THINKING TO RETRIEVE STEP BY STEP

In this section, we introduce DeepRAG, which frames question decomposition, atomic decisions, and answer generation as a Markov Decision Process. Given supervised datasets, we first apply binary tree search to synthesize reasoning data for imitation learning, enabling the model to acquire effective retrieval patterns. We then either employ binary tree search to generate preference data for fine-grained optimization, or use outcome-based rewards to calibrate the LLM's awareness of its knowledge boundaries. The following subsections detail each component of DeepRAG.

### 3.1 OVERVIEW OF THE MDP MODELING

We formalize the step-by-step reasoning process as a Markov Decision Process (Sutton & Barto, 2018), represented by the tuple $(\mathcal{S}, \mathcal{A}, P, R)$, where $\mathcal{S}$ denotes the set of states, $\mathcal{A}$ represents the set of actions, $P$ defines the transition dynamics, and $R$ specifies the reward function.

**States.**  At each step $t$, the state $s_t \in \mathcal{S}$ represents the partial solution to the original question. We denote $s_t = \big[x, (q_1, r_1), \ldots, (q_t, r_t)\big]$, where $x$ is the input question, $q_i$ refers to the $i$-th subquery, and $r_i$ refers to the $i$-th intermediate answer (and any retrieved documents based on $q_i$).

**Actions.** At state $s_t$, the model selects an action $a_{t+1} = (\sigma_{t+1}, \delta_{t+1}) \in \mathcal{A}$, which consists of two sub-decisions:

1. *Termination decision*: Given the partial solution $s_t$, the model makes a binary decision $\sigma_{t+1} \in \{\texttt{continue}, \texttt{terminate}\}$ to determine whether to proceed with generating the next subquery $q_{t+1}$ or finalize the answer $o$.

2. *Atomic decision*: For each subquery $q_{t+1}$, the model decides whether to retrieve external knowledge or rely solely on its parametric knowledge. Formally, this decision is represented as $\delta_{t+1} \in \{\texttt{retrieve}, \texttt{parametric}\}$.

**Transitions.** After executing the action $a_{t+1} = (\sigma_{t+1}, \delta_{t+1})$ in state $s_t$, the environment updates the state to $s_{t+1}$ based on transition dynamics $P$.

Specifically, if $\sigma_{t+1} = \texttt{terminate}$, the process concludes by generating the final answer $o$, resulting in the terminal state $s_{t+1} = \big[x, (q_1, r_1), \ldots, (q_t, r_t), o\big]$. Otherwise, it generates the next subquery $q_{t+1}$. If $\delta_{t+1} = \texttt{retrieve}$, the model retrieves documents $d_{t+1}$ and generates an intermediate answer $ia_{t+1}$ for subquery $q_{t+1}$. Otherwise, it relies on parametric knowledge to generate the intermediate answer. The response $r_{t+1}$ is set as $[d_{t+1}, ia_{t+1}]$ (if retrieved) or $ia_{t+1}$ (if not). The updated state is $s_{t+1} = \big[x, (q_1, r_1), \ldots, (q_{t+1}, r_{t+1})\big]$.

**Rewards.** The reward function evaluates the state based on answer correctness and retrieval cost, applied only after generating the final answer $o$. Formally, $R\big(s_{t+1} = s_t + [o]\big) = -C(o) \times T(s_t)$, where $C(o)$ indicates correctness (1 if correct, $\infty$ otherwise), and $T(s_t)$ represents the total retrieval cost in state $s_t$. Therefore, this reward prioritizes answer correctness while encouraging the model to reduce retrieval cost as much as possible.

## 3.2 BINARY TREE SEARCH

> **Answer Format**
>
> **Question**: <Question>
> **Follow up**: <Subquery1>
> Let's search the question in Wikipedia.
> Context: <Paragraph Text>
> **Intermediate answer**: <Intermediate Answer1>
> **Follow up**: <Subquery2>
> **Intermediate answer**: <Intermediate Answer2>
> ......
> **So the final answer is**: <Answer>

Building on this formulation, LLM iteratively decomposes a given question into subqueries, each derived from previously acquired information. The detailed generation instruction is outlined in Appendix A.1, with the answer format in the left.

Then, we implement a binary tree search to construct reasoning paths that integrate different retrieval strategies for each subquery. As illustrated in Figure 2, given a question, the model generates the $i$-th subquery and explores two answering strategies: directly leveraging parametric knowledge (blue node) or retrieving external documents (green node). Therefore, we can construct a binary tree for each subquery related to the given question, exploring paths based on either parametric knowledge or external knowledge.

## 3.3 IMITATION LEARNING

We present an algorithm that leverages binary trees to identify the optimal reasoning process that leads to the correct final answer while minimizing retrieval costs, corresponding to the highest reward as defined in Section 3.1. Based on the synthesized optimal reasoning data, we fine-tune the model to improve its termination and atomic decisions while enhancing its query decomposition capabilities and generating faithful intermediate answers. We term the resulting model DeepRAG-Imi.

**Synthesizing Data** As shown in Alg. 1, we employ a priority queue to maintain reasoning trajectories based on their retrieval costs. This allows us to efficiently explore potential reasoning paths by iteratively constructing and evaluating them until either finding a correct answer or exhausting all viable options within specified constraints. For instances where no correct answer can be obtained after exhausting all options, we discard them.

Through the synthesis process above, the training dataset obtained contains an adaptive reasoning process, which can be used to facilitate arbitrary LLMs in enhancing the RAG capabilities.

---

**Algorithm 1** Data Construction for Stage I

---

**Require:** Question $x$, answer $y$, language model $\mathcal{M}$, Retriever $\mathcal{R}$, max history length $T$
**Ensure:** Optimal reasoning process $s^*$ or $null$
 1: Initialize priority queue $\mathcal{PQ} \leftarrow \{([x], 0)\}$         ▷ (trajectory, retrieval count)
 2: **while** $\mathcal{PQ}$ is not empty **do**
 3:     $(h, r) \leftarrow \mathcal{PQ}.dequeue()$         ▷ Get trajectory with lowest retrieval count
 4:     $q \leftarrow \mathcal{M}(h)$         ▷ Subquery Generation
 5:     **if** $ShouldAnswer(q)$ or $length(h) > T$ **then**
 6:        $o \leftarrow \mathcal{M}(h, q)$         ▷ Final answer
 7:        **if** $IsEqual(o, y)$ **then return** $h$
 8:     **else**
 9:        $a \leftarrow \mathcal{M}(h, q); \mathcal{PQ}.enqueue(([h, (q, a)], r))$         ▷ Direct answer
10:        $d \leftarrow \mathcal{R}(q)$         ▷ Retrieve document
11:        $a \leftarrow \mathcal{M}(h, q, d); \mathcal{PQ}.enqueue(([h, (q, (d, a))], r + 1))$         ▷ Retrieved answer
12: **return** $null$

---

**Training Objective** We implement a masked loss function for the retrieved documents to prevent the model from learning irrelevant or noisy text that could negatively impact its performance. In this way, we hope the model to enhance the ability to decompose subqueries and retrieve them based on demand. For each instance, the loss function is formulated as follows:

$$\mathcal{L} = -\sum_{1 \leq i \leq n} \log \left[ \Pr(q_i | s_{i-1}) + \Pr(a_i | s_{i-1}, q_i, d_i) \right] \tag{1}$$

where, $d_i$ refers to *null* if there is no reieval for $i$th reasoning step, $n$ refers to the total iteration.

### 3.4 CHAIN OF CALIBRATION

Building on the markov process in Section 3.1, we identify four key optimization aspects for Deep-RAG: termination and atomic decisions, query decomposition, and intermediate answer generation. Unlike the others, *atomic decisions* require the model to recognize its own knowledge boundaries to make precise judgments. Therefore, we propose two calibration variants tailored for different learning paradigms: offline calibration and online calibration.

#### 3.4.1 OFFLINE CALIBRATION

Our approach consists of two key components: (1) synthesizing preference data to determine when retrieval is necessary, and (2) fine-tuning the LLM with this data to enhance its ability to make informed atomic decisions. We term the resulting model DeepRAG-RL$_{off}$.

**Synthesizing Preference Data** First, we identify an optimal path with minimal retrieval based on Alg. 1 using the model trained in Stage I. This provides the optimal atomic decision for each subquery, determining whether retrieval is necessary. From this path, we construct preference pairs for each subquery to indicate the preferred retrieval choice. For example, in Figure 2, the optimal path may suggest answering the first subquery using parametric knowledge while requiring document retrieval for the second. Accordingly, we generate preference pairs favoring parametric knowledge for the first subquery and retrieval for the second. This process enables LLMs to learn when to retrieve external information, thereby improving its ability to maximize the use of parametric knowledge and reducing unnecessary retrievals.

**Learning Objective** We fine-tune the LLM using a learning objective based on the variant of direct preference optimization (Rafailov et al., 2023). Given the $i$-th subquery and the state $s_i = [x, q_1, r_1, \cdots, q_{i-1}, r_{i-1}]$, we have two distinct intermediate answer $r_i^1 = a_i^1$ and $r_i^2 = (d_i, a_i^2)$. Based on the process above, we have known which $r_i$ is preferred. As a result, the training objective can be formulated as follows:

$$\mathcal{L} = -\log \sigma \left( \beta \log \frac{\pi_\theta(y_w \mid s_i, q_i)}{\pi_{\text{ref}}(y_w \mid s_i, q_i)} - \beta \log \frac{\pi_\theta(y_l \mid s_i, q_i)}{\pi_{\text{ref}}(y_l \mid s_i, q_i)} \right) \tag{2}$$

where $\sigma$ is the logistic function, the hyperparameter $\beta$ regulates the penalty imposed for the deviations from the base reference model $\pi_{ref}$. The terms $y_w$ and $y_l$ refer to the generated snippets for

direct answers and retrieved answers, respectively. Specifically, the snippet "Intermediate Answer:" corresponds to a direct answer, while the snippet "Let's search the question on Wikipedia" corresponds to retrieval-based answers.

### 3.4.2 ONLINE CALIBRATION

Online calibration aims to refine the model's knowledge boundaries through outcome-based rewards and self-exploration mechanisms. We term the resulting model DeepRAG-RL$_{on}$.

**Over-retrieve reward shaping**   To steer the model towards achieving both accuracy and efficiency in retrieval-augmented reasoning, we craft a reward function that strikes a balance between answer correctness and retrieval efficiency. Inspired by the overlong reward shaping concept in Yu et al. (2025), we introduce over-retrieve reward shaping. Specifically, we set a maximum retrieval threshold $T$, guiding the model to minimize retrieval steps while still producing correct answers. For a completed state $s_{t+1}$, the reward is defined as:

$$R = \begin{cases} 0, & \text{format}(s_{t+1}) = 0 \\ score_{\text{format}}, & C(o, y) = 0 \ \wedge \ \text{format}(s_{t+1}) = 1 \\ 1 - \alpha \times \min(T, T(s_{t+1})), & C(o, y) = 1 \end{cases} \tag{3}$$

Here, $C(o, y)$ denotes answer correctness, $\text{format}(s_{t+1})$ indicates whether the output format is satisfied, $score_{\text{format}}$ is a reward for correct formatting, $\alpha$ controls the retrieval penalty strength, $T$ is the maximum retrieval budget, and $T(s_{t+1})$ represents the actual retrieval count at state $s_{t+1}$. This design prioritizes correctness, rewards proper formatting, and penalizes excessive retrieval, while capping the penalty at $T$.

**Learning Objectvie**   We employ GRPO (Guo et al., 2025) as the learning objective in DeepRAG-RL$_{on}$. To mitigate the influence of irrelevant or noisy retrieved content, we apply a masked loss over the retrieved tokens, ensuring that the model focuses on learning from informative text. Details are shown in Appendix B.3.

## 4 EXPERIMENT

### 4.1 BASELINES

We use the following baselines to evaluate the performance: **CoT** (Wei et al., 2022) and **CoT\***, which employ 8-shot examples extracted from the training dataset. The asterisk (*) indicates the model output was trained using the same data employed for training the DeepRAG. **CoT-Retrieve** and **CoT-Retrieve\*** augment the eight examples in the context with retrieved relevant documents based on the query. **IterDRAG** (Yue et al., 2024) refers to decomposing the question and answer step by step based on in-context learning. **AutoRAG** (Yu et al., 2024) uses trained models to iteratively decompose questions and retrieve relevant documents for answering. **Search-o1** (Li et al., 2025) leverages special tokens to prompt reasoning models to autonomously invoke retrieval as needed. **Search-R1**(Jin et al., 2025) is trained using reinforcement learning with exact match accuracy as the reward signal for optimizing retrieval decisions. **UAR** (Cheng et al., 2024) employs a trained classifier to determine when to retrieve. **FLARE** (Jiang et al., 2023) and **DRAGIN** (Su et al., 2024) are confidence-based method that decide the timing of retrieval based on token importance and uncertainty. **TAARE** (Zhang et al., 2024) allows the LLM itself to determine when retrieval is needed. **R1-Searcher++**(Song et al., 2025b) employs a composite reward function combining exact match accuracy, format compliance, and group-wise retrieval efficiency for reinforcement learning-based optimization.

### 4.2 DATASETS AND IMPLEMENTATION DETAILS

We use five open-domain QA datasets for our experiments. We treat training datasets as *in-distribution*, and unseen ones as *out-of-distribution*. The in-distribution datasets include HotpotQA (Yang et al., 2018), and 2WikMultihopQA (Ho et al., 2020), and the out-of-distribution datasets consist of

Table 1: The overall experimental results of DeepRAG and other baselines on five benchmarks. The best/second best scores in each dataset are **bolded**/underlined.

| Types | Methods | in-distribution | | | | out-of-distribution | | | | | | Avg |
|---|---|---|---|---|---|---|---|---|---|---|---|---|
| | | Hotpot QA | | 2WikiMultihopQA | | PopQA | | Web Question | | MuSiQue | | |
| | | EM | F1 | EM | F1 | EM | F1 | EM | F1 | EM | F1 | |
| | | | | | | *Llama-3-8B* | | | | | | |
| Reasoning | CoT | 27.20 | 37.75 | 28.20 | 34.85 | 21.20 | 25.33 | 25.20 | 40.56 | 13.70 | 22.97 | 27.69 |
| | CoT-Retreive | 34.90 | 46.85 | 35.80 | 43.41 | 32.80 | 45.87 | 22.90 | 39.22 | 19.10 | 28.18 | 34.90 |
| | CoT* | 21.80 | 31.69 | 25.60 | 30.89 | 23.10 | 25.31 | 26.80 | 40.20 | 4.80 | 13.85 | 24.40 |
| | CoT-Retrieve* | 22.50 | 32.15 | 23.70 | 29.21 | 38.70 | 45.64 | 17.60 | 29.20 | 5.70 | 11.60 | 25.60 |
| | IterDRAG* | 23.20 | 30.95 | 19.60 | 24.80 | 22.70 | 34.53 | 15.90 | 26.79 | 12.40 | 17.75 | 22.86 |
| | Auto-RAG | 25.80 | 36.09 | 23.00 | 30.09 | 27.80 | 42.02 | 17.40 | 32.94 | 19.10 | 28.33 | 28.26 |
| | Search-o1 | 14.80 | 24.08 | 22.20 | 27.10 | 10.30 | 13.54 | 15.30 | 29.60 | 5.40 | 11.98 | 17.43 |
| | Search-R1 | 31.90 | 42.99 | 42.30 | 48.21 | 40.30 | 43.58 | 27.70 | 43.63 | 17.50 | 27.31 | 36.54 |
| Adaptive | FLARE | 23.80 | 32.88 | 30.30 | 37.45 | 28.80 | 40.61 | 28.80 | 40.61 | 14.50 | 23.57 | 30.13 |
| | DRAGIN | 27.60 | 38.05 | 29.10 | 35.68 | 22.60 | 28.53 | 21.20 | 38.72 | 11.80 | 19.97 | 27.33 |
| | UAR | 29.70 | 40.66 | 34.80 | 42.40 | 33.00 | 45.95 | 22.70 | 39.10 | 19.10 | 28.38 | 33.58 |
| | TAARE | 30.60 | 41.43 | 35.20 | 42.85 | 33.20 | 46.01 | 23.40 | 39.56 | 18.60 | 27.55 | 33.84 |
| | R1-Searcher++[†] | 38.90 | 50.27 | 46.20 | 51.78 | 46.10 | 49.14 | 28.20 | 43.57 | 22.80 | 31.56 | 40.85 |
| ours | DeepRAG-Imi | 35.10 | 46.59 | 47.20 | 52.33 | 43.60 | 48.50 | 30.00 | 41.76 | 22.30 | 30.46 | 39.78 |
| | DeepRAG-RL$_{off}$ | 40.70 | 51.54 | 48.10 | 53.25 | 42.50 | 47.80 | 32.70 | 45.24 | 22.50 | 30.40 | 41.47 |
| | DeepRAG-RL$_{on}$ | 37.40 | 48.91 | 49.90 | 55.20 | 47.20 | 50.16 | 30.00 | 44.67 | 25.70 | 34.44 | 42.36 |
| | | | | | | *Qwen-2.5-32B* | | | | | | |
| Reasoning | CoT | 20.60 | 30.62 | 24.40 | 30.94 | 10.90 | 14.45 | 9.70 | 26.00 | 9.50 | 18.26 | 19.54 |
| | CoT-Retreive | 28.60 | 39.43 | 27.90 | 36.73 | 33.80 | 45.91 | 17.20 | 34.15 | 12.90 | 21.98 | 29.86 |
| | IterDRAG | 22.90 | 38.26 | 19.60 | 35.70 | 20.30 | 33.20 | 13.30 | 27.57 | 17.60 | 27.80 | 25.62 |
| | Search-o1 | 34.00 | 45.64 | 29.10 | 35.12 | 23.10 | 30.69 | 17.90 | 35.11 | 16.40 | 25.60 | 29.27 |
| | Search-R1 | 27.00 | 36.42 | 32.30 | 37.01 | 19.00 | 20.16 | 29.10 | 43.96 | 12.50 | 21.10 | 27.85 |
| Adaptive | UAR | 32.10 | 42.32 | 29.00 | 35.90 | 33.70 | 45.75 | 17.00 | 33.92 | 12.80 | 21.80 | 30.43 |
| | TAARE | 31.90 | 42.40 | 29.10 | 35.85 | 33.70 | 45.42 | 12.70 | 28.70 | 13.20 | 22.12 | 29.51 |
| | R1-Searcher++[†] | 34.40 | 45.24 | 45.80 | 49.81 | 42.20 | 45.65 | 24.20 | 41.13 | 21.30 | 29.72 | 37.94 |
| ours | DeepRAG-Imi | 30.40 | 39.44 | 32.00 | 38.32 | 37.50 | 40.72 | 23.90 | 38.62 | 16.50 | 24.67 | 32.21 |
| | DeepRAG-RL$_{off}$ | 32.10 | 41.14 | 40.40 | 44.87 | 40.60 | 43.19 | 24.20 | 38.83 | 19.50 | 32.35 | 35.72 |
| | DeepRAG-RL$_{on}$ | 39.10 | 49.51 | 46.50 | 51.32 | 44.60 | 47.57 | 31.20 | 46.06 | 22.80 | 31.79 | 41.05 |

PopQA (Mallen et al., 2022), WebQuestions (Berant et al., 2013), and MuSiQue (Trivedi et al., 2022). Furthermore, WebQuestions is built upon Freebase to assess model robustness when information may be absent from the knowledge base.

We train our target model on two QA datasets: HotpotQA and 2WikiMultihopQA. For imitation learning, we randomly sample 4,000 examples from each dataset. To enhance the model's question decomposition and context-based generation capabilities, we employ Qwen-2.5-72B to generate the gray (query decomposition) and green nodes (retrieved answers) in Figure 2, and use the target model to generate the blue nodes (parametric answers) for data synthesis. For chain of calibration, we sample an additional 1,000 examples from each dataset. For DeepRAG-RL$_{on}$, the hyperparameters are configured as $score_{\text{format}} = 0.1$, $\alpha = 0.1$, and $T = 5$. The performance is evaluated using Exact Match (EM) and F1 score.

### 4.3 OVERALL RESULTS

Results in Table 1 show DeepRAG's superior performance and robustness across different scenarios.

**DeepRAG demonstrates superior performance across most datasets via thinking to retrieve step by step.** Our method consistently outperforms existing approaches across various backbones and model sizes. Compared to reasoning-based and adaptive RAG baselines, DeepRAG outperforms across all datasets, demonstrating the effectiveness of the structured *retrieval narrative* and on-demand *atomic decisions*. Specifically, the limited performance of IterDRAG highlights the necessity of learning both query decomposition and faithful answering. Confidence-based methods like FLARE struggle to determine the optimal retrieval timing due to their reliance on unstable, predefined metrics. Moreover, we observe that confidence-based methods suffer from instability, as their performance is highly sensitive to threshold selection. Meanwhile, iterative retrieval methods like Auto-RAG often fall into continuous retrieval loops when no highly relevant information is found.

**DeepRAG exhibits remarkable generalization capabilities and robustness in out-of-distribution settings.** DeepRAG-RL$_{off}$ achieves F1 score improvements of 2.63 and 4.57 on PopQA and WebQuestions respectively, even in scenarios where relevant information may be sparse or missing from the knowledge base. Compared to DeepRAG-Imi, DeepRAG-RL$_{on}$ also exhibits superior

Table 2: WebQuestions retrieval frequency: Avg. Retrievals and Time(sec/item).

| Method | EM | Avg. Retrievals | | | Time |
| | | All | Correct | Incorrect | |
|---|---|---|---|---|---|
| FLARE | 28.80 | 0.00 | 0.00 | 0.00 | 2.58 |
| DRAGIN | 21.20 | 0.00 | 0.00 | 0.00 | 1.36 |
| UAR | 22.70 | 0.96 | 0.95 | 0.97 | 0.43 |
| TAARE | 23.40 | 0.66 | 0.65 | 0.66 | 0.11 |
| IterDRAG | 15.90 | 2.25 | 2.16 | 2.27 | 1.09 |
| Auto-RAG | 17.40 | 4.52 | 3.03 | 2.35 | 0.71 |
| R1-Searcher++ | 28.20 | 0.16 | 0.09 | 0.18 | 0.48 |
| Search-R1 | 27.70 | 0.51 | 0.75 | 0.68 | 0.64 |
| DeepRAG-Imi | 30.00 | 0.43 | 0.13 | 0.56 | 0.67 |
| DeepRAG-RL$_{off}$ | 32.70 | 0.28 | 0.12 | 0.36 | 0.50 |
| DeepRAG-RL$_{on}$ | 30.00 | 0.14 | 0.09 | 0.16 | 0.40 |

Table 3: Internal knowledge analysis on 2Wiki-MultihopQA across adaptive retrieval methods.

| Method | F1 | Acc | Balanced Acc | MCC |
|---|---|---|---|---|
| FLARE | 0.000 | 0.718 | 0.500 | 0.000 |
| DRAGIN | 0.007 | 0.709 | 0.495 | -0.045 |
| UAR | 0.481 | **0.756** | 0.648 | 0.341 |
| TAARE | 0.127 | 0.712 | 0.518 | 0.078 |
| Iter-DRAG | 0.000 | 0.718 | 0.500 | 0.000 |
| Auto-RAG | 0.000 | 0.718 | 0.500 | 0.000 |
| R1-Searcher++ | 0.627 | 0.734 | 0.752 | 0.458 |
| Search-R1 | 0.569 | 0.633 | 0.701 | 0.366 |
| DeepRAG-Imi | 0.580 | 0.732 | 0.709 | 0.393 |
| DeepRAG-RL$_{off}$ | 0.621 | 0.749 | 0.743 | 0.451 |
| DeepRAG-RL$_{on}$ | **0.676** | **0.756** | **0.801** | **0.543** |

generalization, with average gains of 2.85 across three out-of-distribution datasets, versus 0.91 for DeepRAG-RL$_{off}$.

**All DeepRAG models effectively explore knowledge boundaries while minimizing hallucination risks.** TAARE often underperforms direct retrieval methods, reflecting a mismatch between the model's internal knowledge and verbose reasoning. Moreover, aggressive fine-tuning strategies such as CoT* and CoT-Retrieve* degrade performance by forcing the model to absorb knowledge beyond its natural boundaries. In contrast, the data-synthesis approach of DeepRAG-Imi enables the model to initially learn on-demand retrieval, while the subsequent DeepRAG-RL$_{off}$ and DeepRAG-RL$_{on}$ further strengthen its ability to make accurate retrieval decisions and refine knowledge utilization.

## 5 ANALYSIS

### 5.1 RETRIEVAL EFFICIENCY

To evaluate the efficiency of our method, we compare the average number of retrievals on the WebQuestions dataset and report the average computation time per query. The computation time is measured on an H20*8 machine. As shown in Table 2, We have the following observations: 1) DeepRAG-RL$_{off}$ and DeepRAG-RL$_{on}$ can achieve higher accuracy with relatively lower retrieval costs, attributed to its dynamic usage of internal knowledge. 2) Confidence-based approaches demonstrate limited robustness across datasets. For instance, neither FLARE nor DRAGIN triggers retrieval under the default confidence threshold in the WebQuestions dataset. 3) Iterative retrieval-based methods typically require numerous retrieval operations. Therefore, efficient adaptive retrieval methods like DeepRAG become crucial for optimizing resource utilization while maintaining performance.

### 5.2 RELEVANCE TO PARAMETRIC KNOWLEDGE

In this section, we investigate the relationship between retrieval needs and internal knowledge to demonstrate how effectively *atomic decisions* explores the knowledge boundary.

Ideally, models should initiate retrieval for queries beyond their parametric knowledge while utilizing their existing knowledge for familiar queries. We use CoT results as an indicator of whether the model can answer questions using its parametric knowledge. Then, we analyze whether other adaptive retrieval methods align with this pattern of parametric knowledge utilization.

We report four metrics. F1 score and Accuracy serve as basic performance measures, while balanced accuracy and Matthews Correlation Coefficient(MCC) (contributors, 2025) are employed to account for the class imbalance between retrieval-required and retrieval-not-required cases.

As shown in Table 3, we find that: 1) DeepRAG demonstrates superior relevance performance across F1, balanced accuracy, and MCC metrics. This suggests that DeepRAG successfully identifies retrieval necessity by exploring knowledge boundary; 2) While FLARE, DRAGIN, and TAARE exhibit high accuracy scores, their relatively low balanced accuracy and MCC scores suggest they mainly succeed in retrieval-required cases but struggle to properly avoid unnecessary retrievals.

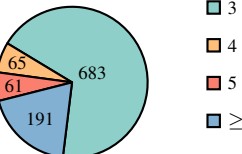 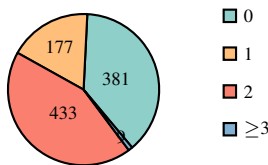 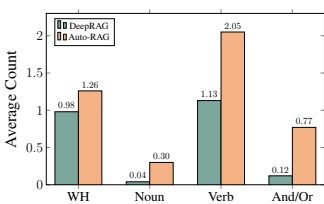

Figure 3: Statistical distributions of the number of subqueries, reflecting how questions are decomposed (left), and the number of retrieval attempts per query (right).

Figure 4: Average counts of WH-words, nouns, verbs, and conjunctions (and/or) per subquery.

## 5.3 QUESTION DECOMPOSITION EFFECTIVENESS

We systematically analyze the effectiveness of question decomposition in *retrieval narrative*. As shown in Figure 3, we present the distribution of subquery counts and retrieval attempts for different questions. Most questions require 3-5 decomposition steps, while retrieval attempts are primarily concentrated within 0-2 rounds. This demonstrates that DeepRAG effectively decomposes questions while minimizing redundant retrieval.

Moreover, we analyze the average counts of WH-words, nouns, verbs, and conjunctions in subqueries, as shown in Figure 4. DeepRAG decomposes atomic queries with fewer pronouns and conjunctions, indicating its concise and effective query decomposition strategy.

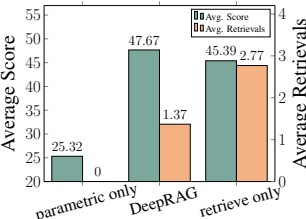 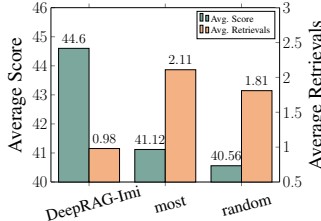 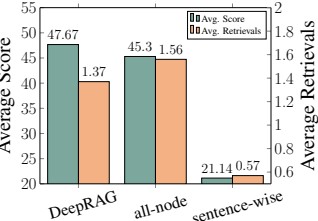

Figure 5: Comparative analysis of retrieval strategies: parametric only or retrieve only.

Figure 6: Average score and retrievals on the ablation study for Imitation Learning.

Figure 7: Average score and retrievals on the ablation study for Chain of Calibration.

## 5.4 DIFFERENT INFERENCE STRATEGY

To gain a deep insight into the effectiveness of *atomic decision*, we evaluate DeepRAG's performance under two extreme scenarios: relying solely on internal knowledge (retrieve only) and using retrieval in each subquery (parametric only). As shown in Figure 5, parametric only yields poor performance, while retrieve only achieves relatively higher accuracy but incurs substantial retrieval costs. DeepRAG achieves superior performance by adaptively selecting between internal and external knowledge sources. Moreover, DeepRAG outperforms the retrieve only approach because retrieval can hinder model performance due to long context or irrelevant knowledge in certain scenarios.

## 5.5 ABLATION STUDY

In this section, we conducted experiments to validate the effectiveness of DeepRAG's data construction and training process.

For Imitation Learning, we compare our default strategy of selecting paths with minimal retrieval cost against two alternative approaches: maximum retrieval cost (*most*) and random path selection (*random*). As shown in Figure 6, DeepRAG-Imi achieves lower retrieval costs and higher average performance compared to both the *most* and *random* methods.

For Chain of Calibration, we compare our default approach of constructing preferences based on nodes from optimal paths against two alternatives: constructing pairs for all nodes and constructing

sentence-level partial order pairs based on retrieval efficiency. As shown in Figure 7, DeepRAG achieves lower retrieval costs while maintaining higher average performance. In contrast, the sentence-level partial order pairs learned incorrect preferences, resulting in over-reliance on internal knowledge and consequently leading to both low retrieval costs and poor performance.

## 6    CONCLUSION

In this paper, we present DeepRAG to model retrieval-augmented reasoning as a Markov Decision Process, enabling strategic and adaptive retrieval by decomposing queries into subqueries and retrieval on demand. Specifically, we develop a binary tree search method to synthesize data for imitation learning and further chain of calibration to train the model in an end-to-end manner. Experiments across various QA tasks show that DeepRAG improves retrieval efficiency while improving answer accuracy by 25.4%, demonstrating its effectiveness in optimizing retrieval-augmented reasoning.

### ETHICS STATEMENT

All datasets and models used in this study are publicly available and have been used in accordance with their respective licenses and terms of use.

### REPRODUCIBILITY STATEMENT

We have clarified our experiment setting in Section 4 and Appendix B.4. We will upload the code to confirm reproducibility, and we promise to open-source the code in the future

### LLM USAGE

Large Language Models (LLMs) were used to aid in refining the language and improving the readability of a limited number of paragraphs in the manuscript.

### ACKNOWLEDGMENTS

We sincerely thank the reviewers for their insightful comments and valuable suggestions. This work was supported by Beijing Natural Science Foundation (L243006), Natural Science Foundation of China (No. 62536008, 62476265, 62306303).

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

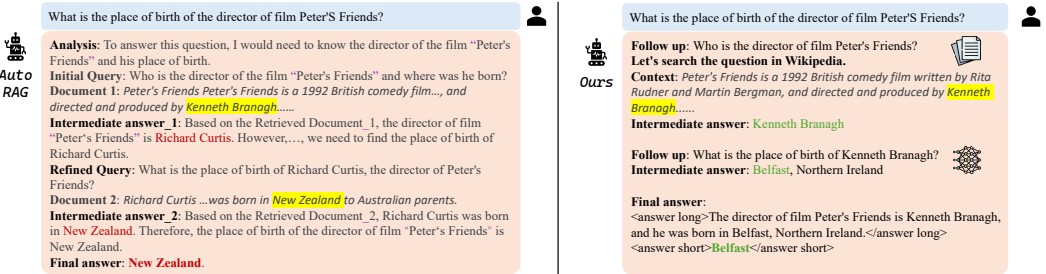

Figure 8: Case Study: Auto-RAG vs. DeepRAG. DeepRAG achieves success by atomic query decomposition, faithful intermediate answer, and adaptively using internal knowledge.

## A TEMPLATES

### A.1 DEEPRAG CONSTRUCT INSTRUCTION

> Instruction: You are a helpful Retrieval-Augmented Generation (RAG) model. Your task is to answer questions by logically decomposing them into clear sub-questions and iteratively addressing each one.
>
> Use "Follow up:" to introduce each sub-question and "Intermediate answer:" to provide answers.
>
> For each sub-question, decide whether you can provide a direct answer or if additional information is required. If additional information is needed, state, "Let's search the question in Wikipedia." and then use the retrieved information to respond comprehensively. If a direct answer is possible, provide it immediately without searching.

## B METHOD DETAILS

### B.1 IMITATION LEARNING OBJECTIVE

We implement a masked loss function for the retrieved documents to prevent the model from learning irrelevant or noisy text that could negatively impact its performance. In this way, we hope the model to enhance the ability to decompose subqueries and retrieve them based on demand. For each instance, the loss function is formulated as follows:

$$\mathcal{L} = - \sum_{1 \leq i \leq n} \log \left[ \Pr(q_i | s_{i-1}) + \Pr(a_i | s_{i-1}, q_i, d_i) \right]$$

where, $d_i$ refers to *null* if there is no reieval for $i$th reasoning step, $n$ refers to the total iteration.

### B.2 OFFLINE LEARNING OBJECTIVE

Given the $i$-th subquery and the state $s_i = [x, q_1, r_1, \cdots, q_{i-1}, r_{i-1}]$, we have two distince intermediate answer $r_i^1 = a_i^1$ and $r_i^2 = (d_i, a_i^2)$. Based on the process above, we have known which $r_i$ is preferred. As a result, the training objective can be formulated as follows:

$$\mathcal{L} = -\log \sigma \left( \beta \log \frac{\pi_\theta(y_w \mid s_i, q_i)}{\pi_{\text{ref}}(y_w \mid s_i, q_i)} - \beta \log \frac{\pi_\theta(y_l \mid s_i, q_i)}{\pi_{\text{ref}}(y_l \mid s_i, q_i)} \right)$$

where $\sigma$ is the logistic function, the hyperparameter $\beta$ regulates the penalty imposed for the deviations from the base reference model $\pi_{ref}$. The terms $y_w$ and $y_l$ refer to the generated snippets for direct answers and retrieved answers, respectively. Specifically, the snippet "Intermediate Answer:" corresponds to a direct answer, while the snippet "Let's search the question on Wikipedia" corresponds to retrieval-based answers.

### B.3 ONLINE LEARNING OBJECTIVE

**Learning Objectvie** We employ GRPO (Guo et al., 2025) as the learning objective in DeepRAG-$\text{RL}_{on}$. To mitigate the influence of irrelevant or noisy retrieved content, we apply a masked loss over the retrieved tokens, ensuring that the model focuses on learning from informative text.

$$
\begin{aligned}
\mathcal{J}_{\text{GRPO}}(\theta) = \mathbb{E}_{q \sim P(Q), \{o_i\}_{i=1}^{G} \sim \pi_{\theta_{\text{old}}}(\cdot|q)} \Bigg[ \\
\frac{1}{\sum_{i=1}^{G}|o_i|} \sum_{i=1}^{G} \sum_{t=1}^{|o_i|} \min\left(r_{i,t}(\theta)\,\hat{A}_{i,t},\ \text{clip}\left(r_{i,t}(\theta), 1-\epsilon, 1+\epsilon\right)\hat{A}_{i,t}\right) * \mathbb{I}(o_{i,t}) - \beta\, D_{\text{KL}}\left(\pi_\theta \,\|\, \pi_{\text{ref}}\right) \Bigg]
\end{aligned}
\tag{4}
$$

,where $\mathbb{I}(o_{i,t})$ is an indicator function that equals 0 if token $o_{i,t}$ belongs to retrieved text, and 1 otherwise. $r_{i,t}$ refers to importance sampling ratio and $\hat{A}_{i,t}$ refers to advantages as shown in Equation B.4.

$$
r_{i,t}(\theta) = \frac{\pi_\theta(o_{i,t} \mid q, o_{i,<t})}{\pi_{\theta_{\text{old}}}(o_{i,t} \mid q, o_{i,<t})}, \qquad \hat{A}_{i,t} = \frac{R_i - \text{mean}(\{R_i\}_{i=1}^{G})}{\text{std}(\{R_i\}_{i=1}^{G})}.
\tag{5}
$$

### B.4 IMPLEMENTATION DETAILS

We train our target model on two QA datasets: HotpotQA and 2WikiMultihopQA. For imitation learning, we randomly sample 4,000 examples from each dataset. To enhance the model's question decomposition and context-based generation capabilities, we employ Qwen-2.5-72B to generate the gray (query decomposition) and green nodes (retrieved answers) in Figure 2, and use the target model to generate the blue nodes (parametric answers) for data synthesis. For chain of calibration, we sample an additional 1,000 examples from each dataset. For DeepRAG-$\text{RL}_{on}$, the hyperparameters are configured as $score_{\text{format}} = 0.1$, $\alpha = 0.1$, and $T = 5$. The performance is evaluated using Exact Match (EM) and F1 score.

Following Su et al. (2024), we adopt BM25 for retrieval and Wikipedia[2] as knowledge base. We selected Llama-3-8B-Instruct (Dubey et al., 2024), and Qwen-2.5-32B (Yang et al., 2024) as our target model. To implement Search-o1, we employ the distillation series of DeepSeek-R1 (Guo et al., 2025), as the method depends on reasoning models. All reinforcement learning experiments are reproduced based on the VeRL framework[3].

## C DETAILED ANALYSIS

### C.1 CASE STUDY

As illustrated in Figure 8, we conduct a case study comparing DeepRAG with Auto-RAG (Yu et al., 2024), a closely related method that utilizes iterative retrieval for retrieval-augmented generation. For each subquery, Auto-RAG retrieves relevant documents and generates a corresponding subanswer. This approach is not only time-consuming but also fails when no relevant documents are retrieved. Although Auto-RAG attempts to address this issue using its own relevant documents, it falls into endless loops in most cases. In contrast, DeepRAG iteratively generates subqueries and determines whether to use internal knowledge at each iteration. The binary tree search data synthesis method for optimization ensures reliable subquery generation, intermediate answers, and final answers. Even when no related information exists in retrieved documents, the model is directed to provide a final answer based on internal knowledge.

### C.2 RELEVANCE TO PARAMETRIC KNOWLEDGE

In this section, we investigate the relationship between retrieval needs and parametric knowledge to demonstrate how effectively our method explores the knowledge boundary.

---

[2]`https://dl.fbaipublicfiles.com/dpr/wikipedia_split/psgs_w100.tsv.gz`
[3]`https://github.com/volcengine/verl`

Ideally, models should initiate retrieval for queries beyond their parametric knowledge while utilizing their existing knowledge for familiar queries. We use CoT results as an indicator of whether the model can answer questions using its parametric knowledge. Subsequently, we analyze whether other adaptive retrieval methods align with this pattern of parametric knowledge utilization. We evaluate the relevance using four metrics. F1 score and Accuracy serve as basic performance measures, while balanced accuracy and Matthews Correlation Coefficient(MCC) are employed to account for the class imbalance between retrieval-required and retrieval-not-required cases. The MCC ranges from -1 to 1, where a value of 1 indicates perfect correlation, 0 represents no correlation (random chance), and -1 signifies an inverse correlation.

As shown in Table 3, we find that 1) DeepRAG demonstrates superior relevance performance across F1, balanced accuracy, and MCC metrics. This suggests that DeepRAG successfully identifies retrieval necessity by exploring knowledge boundary. 2) While FLARE, DRAGIN, and TAARE exhibit high accuracy scores, their relatively low balanced accuracy and MCC scores suggest they mainly succeed in retrieval-required cases but struggle to properly avoid unnecessary retrievals.

## C.3 PERFORMANCE AGAINST STRONG BASELINE MODELS

We compare the Llama-3-8B-based DeepRAG with recent strong reasoning models: QwQ-32B-preview (Team, 2024) and gpt-4o-turbo (OpenAI). As shown in Table 4, DeepRAG achieves superior average performance over QwQ and gpt-4o, particularly in time-sensitive QA tasks. While DeepRAG does not surpass gpt-4o in some cases, it achieves comparable performance levels. These results demonstrate that by adaptively leveraging retrieval, DeepRAG can achieve an equivalent level of factual accuracy to the parametric knowledge of strong reasoning models.

| Models | ID | PopQA | WQ | Avg |
|---|---|---|---|---|
| QwQ-32B | 31.43 | 10.60 | 15.10 | 19.04 |
| gpt-4o-turbo | **60.60** | 43.50 | 25.35 | 43.15 |
| DeepRAG-RL$_{on}$ | 52.06 | **47.20** | **30.00** | **46.09** |

Table 4: Performance against strong baseline models.

