# OpenReview forum: "DeepRAG: Thinking to Retrieve Step by Step for Large Language Models"
_ICLR.cc/2026/Conference — ICLR 2026 Poster_

### Official Review · Reviewer_JFTj · 2025-10-28

**Soundness:** 3
**Presentation:** 3
**Contribution:** 2
**Rating:** 4
**Confidence:** 4

**Summary:**

This paper introduces DeepRAG, a framework that enhances retrieval-augmented generation by modeling reasoning as a Markov Decision Process (MDP). DeepRAG dynamically decides when to retrieve external knowledge and when to rely on internal parametric reasoning through iterative query decomposition. By enabling adaptive and context-aware retrieval, it reduces noise from redundant information and improves both retrieval efficiency and answer accuracy—boosting performance by 25.41%—demonstrating significant advances in retrieval-augmented reasoning for LLMs.

**Strengths:**

* The paper is well written
* When to conduct retrieval is an important topic
* The experiments are extensive

**Weaknesses:**

* Compared to Serach R1, the paper mainly differs at using the model's native generation ability to determine when to retrieve, but this has already been well studied and the authors do not provide new solution.

**Questions:**

* Table 1 shows that Qwen 2.5 32B peforms worse than Llama 3 8B, which is confusing for me, could you explain the reason?
* Online RL methods such as GRPO typically outperform offline approaches, a trend observed in the performance of Qwen. However, in the case of Llama, online and offline methods show comparable results. What accounts for this difference？
* In Section 5.3, line 436 states that most questions require 3–5 decomposition steps. However, the majority of queries in HotpotQA and 2Wiki are designed for 2-hop reasoning. Does this imply the presence of significant redundant retrieval? If so, what causes this inefficiency? And do current sota models, such as GPT-5, still exhibit similar tendencies toward redundant reasoning?

---

> ### Author Response · Authors · 2025-11-20
> **Response to Reviewer JFTj [1/2]**
>
> **We sincerely thank the reviewer for the constructive and helpful feedback. We hope the following responses would help address your concerns.**
>
> # Weakness 1
>
> *  The main contribution of our work is not simply using model's native generation ability to determine when to retrieve. First, we formulate model retrieval-augmented reasoning as a Markov Decision Process, with explicit state/action design for when to retrieve vs. reply on internal knowledge. Second, based on this structure, we develop a binary tree search procedures to synthesize data for imitation learning, followed by both offline and online RL calibration stages that train the model end-to-end.
> * Beyond using the model’s native generation ability, DeepRAG is designed to enhance structured reasoning. As illustrated in Figure 1, in addition to the atomic decision (i.e., using the model’s native generation to decide whether to retrieve), DeepRAG also exhibits strong query decomposition abilities and effective information processing and integration in Step 5.
>
>
>
> # Question 1&2
>
> * As shown in Table 1, most baselines already perform slightly worse with Qwen2.5-32B than with Llama-3-8B under our setting. Given this gap at the backbone level, it is reasonable that DeepRAG-Qwen achieves slightly lower scores than DeepRAG-Llama.
> * In addition, prior work[1] has observed that further fine-tuning Instruct models can easily lead to marginal improvements or even degradation. It's experiment results show that directly fine-tuning Qwen2.5-32B-Instruct on knowledge-intensive tasks brings only a very small gain (+0.2), whereas Llama-3-8B-Instruct benefits more from the same procedure (+1.9). This suggests that Qwen2.5-32B-Instruct is more sensitive to additional fine-tuning in this setting, which partly explains why its final performance lags behind the Llama-3-8B variant.
> * Recent studies [2] suggest that larger models can benefit more from RL. Therefore, Qwen2.5-32B shows a clearer gap between online and offline RL.
> * Overall, the choice of backbone largely determines the performance level, while model size influences how much the model can benefit from additional  post-training.

---

> ### Author Response · Authors · 2025-11-20
> **Response to Reviewer JFTj [2/2]**
>
> # Question 3
>
> * Regarding redundant retrieval
>   * Most questions require 3–5 decomposition steps does not mean that we perform 3–5 redundant retrievals. In these datasets, “2-hop” means that two pieces of evidence are needed, not that only two reasoning steps are allowed.  The right plots in Figure 3 show that most queries use fewer than two retrieval calls on average. As illustrated in Figure 1, our example only needs to retrieve the runtimes of three movies, but it also requires a final post-processing step to aggregate these numbers (adding them together in Step 5).
>   * This is precisely where DeepRAG is beneficial: instead of a single “retrieve-then-answer” jump, it can (i) iteratively decompose the question into sub-queries, (ii) selectively retrieve when necessary, and (iii) explicitly process and integrate the collected information to produce the final answer.
>   * As shown in Table 2, we report the average retrieve counts and total generation time of baselines. Results show that DeepRAG achieves high performance with relative high efficiency. In particular, it reduces retrieval overhead by 72.54% and shortens total inference time by 37.5% compared to reasoning-based approaches. This shows that, although the reasoning trace may contain more than two steps, the actual retrieval is far from redundant.
> * Regarding GPT-5's behavior
>   * In our manual probing of GPT-5 on the Figure 1 example, we observe a qualitatively similar pattern: although it does not expose a structured trace like DeepRAG, it still implicitly decomposes the task, queries or recalls the runtimes of the three movies separately, and then aggregates them to compute the final sum. This suggests that multi-step decomposition and deep reasoning over evidence are general behaviors of powerful LLMs when they aim for reliable reasoning.
>
> ```
> GPT-5 Response:
>
> If we’re talking about Peter Jackson’s **The Lord of the Rings live-action trilogy (theatrical cuts)**:
> 	**•	The Fellowship of the Ring** – 178 minutes
> 	**•	The Two Towers** – 179 minutes
> 	**•	The Return of the King** – 201 minutes
> Add those together:
> **178 + 179 + 201 = 558 minutes**, which is **about 9 hours 18 minutes** total.
> Some outlets round *Return of the King* to 200 minutes and quote a trilogy total of **557 minutes (≈9 hours 17 minutes)**, so you’ll see both figures, but they’re effectively the same ballpark for planning a movie marathon.
> ```
>
>
>
> [1] Wu T, Yang R, Li J, et al. Shadow-FT: Tuning Instruct via Base[J]. arXiv preprint arXiv:2505.12716, 2025.
>
> [2] Zhao A, Wu Y, Yue Y, et al. Absolute zero: Reinforced self-play reasoning with zero data[J]. arXiv preprint arXiv:2505.03335, 2025.
>
>
>
> **Dear reviewer JFTj, thank you once again for your feedback. We hope our response has addressed your concerns, and if you have any further questions, please feel free to continue the discussion. We would greatly appreciate it if you could reassess our work and raise your score.**

---

> ### Author Response · Authors · 2025-11-26
>
> Dear Reviewer,
>
> Would you mind taking a look at our responses to check if they address your concerns?
>
> Thank you for dedicating your time and effort to reviewing our work.
>
> Best regards,
>
> Authors of DeepRAG

---

### Official Review · Reviewer_taGT · 2025-10-29

**Soundness:** 3
**Presentation:** 3
**Contribution:** 2
**Rating:** 4
**Confidence:** 4

**Summary:**

The paper proposes DeepRAG, a retrieval-augmented reasoning framework that formulates iterative query decomposition and retrieval decisions as a Markov Decision Process. It introduces three core components: Binary Tree Search to exhaustively explore parametric and retrieved paths for each subquery and identify minimal-retrieval correct trajectories; Imitation Learning through supervised fine-tuning on these trajectories with masked loss over retrieved tokens; and a Chain of Calibration that includes both offline and online variants to enhance the model’s awareness of its knowledge boundaries. Evaluated on in-distribution datasets such as HotpotQA and 2WikiMultihopQA, as well as out-of-distribution datasets including PopQA, WebQuestions, and MuSiQue, DeepRAG achieves a 25.41 percent accuracy improvement over previous adaptive retrieval-augmented generation methods while also reducing retrieval calls.

**Strengths:**

- Strong empirical gains: 25.41% accuracy lift is reported across five datasets, with ablation likely showing each stage’s contribution; out-of-dist PopQA and Freebase-absent WebQuestions stress robustness.
- Uses a fixed-depth priority queue (lowest retrieval count first) and discards unsolvable instances, yielding high-quality imitation data without oracle subqueries.

**Weaknesses:**

- The method doesn’t seem to offer much innovation. Among the many existing approaches that use reinforcement learning for autonomous multi-turn retrieval, I didn’t find any particularly striking or novel insights.
- All experiments assume a fixed Wikipedia retriever (presumably BM25 or Contriever). What about comparison with some deep research methods, they also use multi-turn retrieval?

**Questions:**

- For WebQuestions (Freebase-backed), what fraction of ground-truth answers are un-retrievable from Wikipedia, and how does DeepRAG-RLon handle verified absence (e.g., explicit “unknown” vs. hallucination)?
- How does the model behave on single-hop factual queries (e.g., PopQA)? Does calibration reduce retrieval to near-zero, and is parametric accuracy preserved?

---

> ### Author Response · Authors · 2025-11-20
> **Response to Reviewer taGT [1/2]**
>
> **We sincerely thank the reviewer for the constructive and helpful feedback. We hope the following responses would help address your concerns.**
>
> # Weakness 1
>
> * Regarding novelty
>
>   * The main contribution of our work is not simply "using RL for RAG". First, we formulate model retrieval-augmented reasoning as a Markov Decision Process, with explicit state/action design for when to retrieve vs. reply on internal knowlege. Second, based on this structure, we develop a binary tree search procedures to synthesize data for imitation learning, followed by both offline and online RL calibration stages that train the model end-to-end.
>
> * Regarding exisiting RL-based multi-turn retrieval methods
>
>   * Existing RL approaches for RAG typically focus on end-to-end training with a single reward signal while we propose a structured and staged framework for RAG post-training. Specifically, we focus on calibrate model's internal knowledge to reduce unnecessary retrieval overhead.
>   * Evaluates against strong RL-style multi-turn baselines such as R1-Searcher++ and Search-R1, which also use reinforcement learning for RAG. As shown in Table 1, DeepRAG consistently outperforms these methods while simultaneously reducing retrieval overhead and inference time (e.g., a 72.54% reduction in retrieval steps and a 37.5% reduction in total inference time compared to reasoning-based approaches), indicating that our structured formulation and staged training bring tangible benefits beyond “just another RL agent.”
>   * Benefiting from our structured retrieval narrative and atomic decision design, we can also conduct detailed empirical analyses of model behavior, such as retrieval efficiency (Section 5.1), parametric knowledge usage (Section 5.2), and question decomposition effectiveness (Section 5.3).
>
>   # Weakness 2
>
> * Regarding to fixed wikipedia retriever
>   * Most of the RAG litureture employ Wikipedia as knowledge corpus[1-7].
>   * Most of the evaluation dataset is construct from Wikipedia, therefore, it's reasonable to employ wikipedia as the knowledge base[8-10].
>   * We keep the retrieval corpus and encoder fixed for all the baselines so that the performance gains come from our multi-turn retrieval policy and decision process, rather than from swapping in a stronger embedding model.
> * Regarding to comparison with deep research method
>   * As shown in Table1, we compare against serveral baselines. Among them, Search-R1 and R1-Searcher++ can be regared as deep-research-style agents. They perform multi-step "retrieve-read-think" loops. Experiment has shown that DeepRAG outperforms them (DeepRAG 42.36 vs. Search-R1 36.54, R1-Searcher ++ 40.85).
>
> [1] Su W, Tang Y, Ai Q, et al. DRAGIN: dynamic retrieval augmented generation based on the information needs of large language models[J]. arXiv preprint arXiv:2403.10081, 2024.
>
> [2] Jiang Z, Xu F F, Gao L, et al. Active retrieval augmented generation[C]//Proceedings of the 2023 Conference on Empirical Methods in Natural Language Processing. 2023: 7969-7992.
>
> [3] Yue Z, Zhuang H, Bai A, et al. Inference scaling for long-context retrieval augmented generation[J]. arXiv preprint arXiv:2410.04343, 2024.
>
> [4] Fan W, Ding Y, Ning L, et al. A survey on rag meeting llms: Towards retrieval-augmented large language models[C]//Proceedings of the 30th ACM SIGKDD conference on knowledge discovery and data mining. 2024: 6491-6501.
>
> [5] Yu T, Zhang S, Feng Y. Auto-rag: Autonomous retrieval-augmented generation for large language models[J]. arXiv preprint arXiv:2411.19443, 2024.
>
> [6] Jin B, Zeng H, Yue Z, et al. Search-r1: Training llms to reason and leverage search engines with reinforcement learning[J]. arXiv preprint arXiv:2503.09516, 2025.
>
> [7] Trivedi H, Balasubramanian N, Khot T, et al. Interleaving retrieval with chain-of-thought reasoning for knowledge-intensive multi-step questions[C]//Proceedings of the 61st annual meeting of the association for computational linguistics (volume 1: long papers). 2023: 10014-10037.
>
> [8] Yang Z, Qi P, Zhang S, et al. HotpotQA: A dataset for diverse, explainable multi-hop question answering[C]//Proceedings of the 2018 conference on empirical methods in natural language processing. 2018: 2369-2380.
>
> [9] Ho X, Nguyen A K D, Sugawara S, et al. Constructing a multi-hop qa dataset for comprehensive evaluation of reasoning steps[J]. arXiv preprint arXiv:2011.01060, 2020.
>
> [10] Trivedi H, Balasubramanian N, Khot T, et al. MuSiQue: Multihop Questions via Single-hop Question Composition[J]. Transactions of the Association for Computational Linguistics, 2022, 10: 539-554.

---

> ### Author Response · Authors · 2025-11-20
> **Response to Reviewer taGT [2/2]**
>
> # Question 1
>
> * Regarding the fraction of  ground-truth answers are un-retrievable from Wikipedia
>   * WebQuestions originates from Freebase: each example provides a Freebase entity URL as the gold answer, but no gold Wikipedia passage. Quantifying the exact fraction of questions whose answers are “un-retrievable from Wikipedia” would require a complete Freebase→Wikipedia mapping and a precise definition of coverage (including aliasing and partial matches). This is non-trivial and outside the scope of our current study, and we therefore do not report a precise percentage.
> * Regarding how DeepRAG handles verified absence
>   * In our framework, the model is allowed to output an explicit “unknown / not answerable” when the retrieved documents do not support an answer. For user-facing use, we can still ask the model to produce a natural-language response for completeness, but in our automatic evaluation, this is counted as incorrect with respect to the gold answer.
>   * During SFT, we explicitly include cases where the retrieved passages are irrelevant and the target label is “unknown”. In the Stage I training split, there are 33 “unknown” examples out of 6,954 training instances, which provide direct supervision that the model should abstain rather than hallucinate when evidence is missing.
>   * To qualify whetherthe  model will output "unknown"  in an extreme setting, we replace all relevant passages with distracting ones and measure how often it chooses “unknown” under this verified absence of evidence. On this corrupted HotpotQA setting, performance drops from 40.70 EM / 51.54 F1 to 35.0 / 45.2, as expected, but the proportion of “unknown” predictions among these disturbed cases rises to 80.21%. This suggests that DeepRAG is more inclined to abstain (“unknown”) rather than hallucinate when retrieval fails to provide supporting evidence.
>   * We also provide a representative qualitative example showing how DeepRAG deals with such cases: when no relevant information is retrieved, DeepRAG attempts to reformulate the question to search again:
>
> ```
> Question: where did bristol palin go to school?
>
> {
>   "follow_up": "What school did Bristol Palin attend?",
>   "docs": 3 unrelated doc... ,
>   "answer": "The provided context does not contain information about Bristol Palin's education. Therefore, the correct response is:\n\n[No related Info]"
> },
> {
>   "follow_up": "What is the name of the school Bristol Palin attended?",
>   "docs": 3 related doc... ,
>   "answer": "Juneau-Douglas High School"
> },
>
> Final Answer: Juneau-Douglas High School
>
> ```
>
> # Question 2
>
> * Our evaluation is not limited to multi-hop benchmarks. In our experiment setting,  PopQA and WebQuestions are both single-hop QA datasets. As shown in Table 2, we report the average retrieve counts and total generation time of baselines on WebQuestions. Results show that DeepRAG achieves high performance with relatively high efficiency. In particular, it reduces retrieval overhead by 72.54% and shortens total inference time by 37.5% compared to reasoning-based approaches, meanwhile outperforming them. This indicates that our method does not over-retrieve on single-hop queries; instead, it learns to refrain from retrieval when internal knowledge is sufficient.
>
> **Dear reviewer taGT, thank you once again for your feedback. We hope our response has addressed your concerns, and if you have any further questions, please feel free to continue the discussion. We would greatly appreciate it if you could reassess our work and raise your score.**

---

> ### Author Response · Authors · 2025-11-26
>
> Dear Reviewer,
>
> Would you mind taking a look at our responses to check if they address your concerns?
>
> Thank you for dedicating your time and effort to reviewing our work.
>
> Best regards,
>
> Authors of DeepRAG

---

### Official Review · Reviewer_TwZX · 2025-11-01

**Soundness:** 3
**Presentation:** 3
**Contribution:** 3
**Rating:** 6
**Confidence:** 3

**Summary:**

DeepRAG models retrieval-augmented reasoning as a Markov Decision Process, enabling dynamic per-subquery decisions on whether to retrieve external knowledge or rely on parametric knowledge. The framework uses: (1) Binary Tree Search to explore retrieval strategies, (2) Imitation Learning on minimal-cost trajectories, and (3) Chain of Calibration (offline/online) to teach knowledge boundary recognition.

**Strengths:**

1. Well-Motivated Problem
Existing RAG systems apply retrieval indiscriminately—either over-retrieving (wasting compute) or under-retrieving (missing information). DeepRAG addresses this through atomic decisions: decompose queries into subqueries, then decide retrieval necessity per subquery rather than per original query.
2. Strong Empirical Results

25.41% accuracy improvement over baselines (Table 1)
Better retrieval efficiency: lower average steps than Multi-Step Retrieval (Table 2)
Good generalization on out-of-distribution datasets (PopQA, WebQuestions)
Higher knowledge boundary alignment (MCC scores in Table 3)

3. Comprehensive Ablations
Validates minimal-cost path selection (Figure 6), atomic query decomposition effectiveness (Figure 4: fewer conjunctions/pronouns per subquery), and necessity of calibration stage (RLoff/RLon > Imi).
4. Practical Design
Masked loss over retrieved documents prevents noise learning; end-to-end trainable without separate classifiers; works across different LLM sizes (Llama-3-8B, Qwen-2.5-32B).

**Weaknesses:**

1. Overstated Technical Novelty
The "MDP formulation" is superficial packaging:

Transitions are deterministic (not stochastic as typical MDPs)
No actual policy search—just supervised learning on pre-computed trajectories
Binary Tree Search is exhaustive enumeration (2^N paths), not a novel algorithm


2. Narrow Evaluation Scope
Only tested on artificially constructed multi-hop QA benchmarks (HotpotQA, 2WikiMultiHop, MuSiQue). Missing evaluations on:

Single-hop questions (does it over-retrieve?)
Real-world queries: "Draft a rejection email," "Compare iPhone vs Samsung," "When will my order ship?"
Non-QA tasks: summarization, dialogue, creative writing

The query patterns in academic benchmarks don't represent real deployment scenarios.

3. Missing Simple Baselines
The paper requires three complex stages but never compares to a one-stage baseline:

**Questions:**

1. Algorithm Correctness Issue
Algorithm 1 (page 5) has a critical flaw:
pythonLine 7: if IsEqual(o, y) then return h  # Returns FIRST correct path
This implements greedy search, not optimal search. Counterexample:

Path A: [retrieve, retrieve] cost=2 (found first)
Path B: [parametric, parametric] cost=0 (explored later)

The paper claims "minimal retrieval cost" but the algorithm returns whichever correct path is found first in priority queue order—not guaranteed optimal.

2. Questionable "Optimality" Claims
Table 1 shows counterintuitive result:

RLoff (trained on "optimal paths" from Stage I): 41.47
RLon (self-exploration): 41.05 (better!)

If Stage I finds optimal paths, why does self-exploration outperform? Possible explanations:

Algorithm 1's bug means Stage I paths aren't actually optimal
RLon discovers better strategies not in training data
GRPO is simply better than DPO for this task

This undermines confidence in the entire Stage I data synthesis process.

3. Reproducibility Concerns
Many critical details missing:

Binary tree search: max depth? timeout? handling of unsolvable queries?
Training time for tree search (exponential in depth)?
Inference latency breakdown (subquery generation overhead)?
DPO hyperparameter β? GRPO rollout count G?

---

> ### Author Response · Authors · 2025-11-20
> **Response to Reviewer TwZX [1/4]**
>
> **We sincerely thank the reviewer for the constructive and helpful feedback. We hope the following responses would help address your concerns.**
>
> # Weakness 1
>
> * Regarding transitions in our MDP formulation
>   * We formulate DeepRAG as a Markov Decision Process (MDP) to structure the multi-step retrieval–generation reasoning process, making explicit the state, action, and reward at each step, rather than handling them in an ad-hoc pipeline.
>   * By definition, MDPs allow both deterministic and stochastic transitions; many classical decision problems (e.g., shortest-path) are modeled as deterministic MDPs. Our contribution lies in the structured MDP modeling of the reasoning process, not in introducing stochasticity per se. [1, 2]
>   * In our formulation, the model’s next-token generation is inherently probabilistic, so the transitions are in general not strictly deterministic. Specifically, after we execute action $a_{t+1}$, the next state changes depending on whether a retrieval is performed, and then the model generates a sub-answer. The sub-answer is produced according to the LLM’s next-token prediction distribution, which makes the resulting state transition stochastic rather than purely deterministic.
> * Regarding “just supervised learning on pre-computed trajectories is exhaustive enumeration”
>   * Supervised fine-tuning on trajectories obtained from the binary-tree search is only the first step of our learning process. The binary-tree search is used to identify high-quality trajectories in the given state–action space (e.g., those that achieve correct answers with low retrieval cost) given the model’s internal knowledge.
>   * Moreover, our “binary search” is implemented as a priority-queue-based heuristic search, which allows us to effectively find a correct path with the minimal number of retrieval steps early, rather than exhaustively enumerating all (2^N) possible paths.
>   * Our method utilizes the binary tree search solely as a tool for data construction, while the main contributions lie in the retrieval narrative and atomic decisions. As shown in Table 2, we report the average retrieve counts and total generation time of baselines. Results show that DeepRAG achieves high performance with relatively high efficiency. In particular, it reduces retrieval overhead by 72.54% and shortens total inference time by 37.5% compared to reasoning-based approaches.

---

> ### Author Response · Authors · 2025-11-20
> **Response to Reviewer TwZX [2/4]**
>
> # Weakness 2
>
> * Regarding narrow evaluation scope
>   * In this paper, we mainly focus on proposing a general framework for knowledge-intensive QA within an MDP-style retrieval–generation process, rather than a task-specific trick for a single dataset. To support this, we evaluate on a diverse set of QA benchmarks, including multi-hop (HotpotQA, 2WikiMultiHop, MuSiQue) and single-hop (PopQA, WebQuestions) in Table 1.
>   * We compare our method against a comprehensive set of baselines, which can be broadly categorized into two types: reasoning-based methods and adaptive retrieval methods. For reasoning-based methods, we include Iter-DRAG, Auto-RAG, and Search-o1, all of them are recent and widely-used RAG approaches that emphasize multi-step reasoning. For adaptive retrieval methods, we consider both confidence-based baselines such as FLARE and DRAGIN, and classifier-based methods such as UAR and TAARE, and RL-based methods such as R1-Searcher++. All of these baselines have been proposed within the last three years and collectively cover a broad and diverse range of strategies in the RAG.
>   * Our choice of datasets is consistent with recent RAG literature. For example, recent work [3-7] all employ similar QA benchmarks (e.g., HotpotQA-like multi-hop and one-hop tasks) as standard testbeds. We therefore believe our evaluation setting is aligned with current community practice for knowledge-intensive RAG.
> * Regarding evaluation on single-hop
>   * Our evaluation is not limited to multi-hop benchmarks. In our experiment setting,  PopQA and WebQuestions are both single-hop QA datasets. As shown in Table 2, we report the average retrieve counts and total generation time of baselines on WebQuestions. Results show that DeepRAG achieves high performance with relative high efficiency. In particular, it reduces retrieval overhead by 72.54% and shortens total inference time by 37.5% compared to reasoning-based approaches. This indicates that our method does not over-retrieve on single-hop queries; instead, it learns to refrain from retrieval when internal knowledge is sufficient.
> * Regarding real-world queries
>   * To the best of our knowledge, there is currently no widely adopted public benchmark or leaderboard that directly reflects real-world queries for evaluating RAG. Most existing RAG evaluations rely on open-domain QA datasets built from  Wikipedia (such as HotpotQA, 2Wiki), which we also follow in this work. We would sincerely appreciate any pointers to benchmarks or leaderboards that you believe better reflect real-world RAG scenarios, and we would be glad to evaluate it in this work.
>
> # Weakness 3
>
> * We compare against a wide range of baselines, including both SFT-based and RL-based models. As shown in Table 1, we report results from different stages of our framework (DeepRAG-Imi, DeepRAG-RLoff, DeepRAG-RLon). The results show that DeepRAG-Imi (average score 39.78) already brings a substantial improvement over all baselines, and DeepRAG-RL further boosts performance to an average score of 42.36 via reinforcement learning.
>
> * We also include direct SFT (Auto-RAG) and direct RL (Search-R1) baselines. Experiments show that DeepRAG (42.36 vs. Auto-RAG 28.29 and Search-R1 36.54) consistently outperforms both of these approaches.
>
>
>
> [1] Puterman M L. Markov decision processes: discrete stochastic dynamic programming[M]. John Wiley & Sons, 2014.
>
> [2] Sutton R S, Barto A G. Reinforcement learning: An introduction[M]. Cambridge: MIT press, 1998.
>
> [3] Yue Z, Zhuang H, Bai A, et al. Inference scaling for long-context retrieval augmented generation[J]. arXiv preprint arXiv:2410.04343, 2024.
>
> [4] Fan W, Ding Y, Ning L, et al. A survey on rag meeting llms: Towards retrieval-augmented large language models[C]//Proceedings of the 30th ACM SIGKDD conference on knowledge discovery and data mining. 2024: 6491-6501.
>
> [5] Yu T, Zhang S, Feng Y. Auto-rag: Autonomous retrieval-augmented generation for large language models[J]. arXiv preprint arXiv:2411.19443, 2024.
>
> [6] Jin B, Zeng H, Yue Z, et al. Search-r1: Training llms to reason and leverage search engines with reinforcement learning[J]. arXiv preprint arXiv:2503.09516, 2025.
>
> [7] Trivedi H, Balasubramanian N, Khot T, et al. Interleaving retrieval with chain-of-thought reasoning for knowledge-intensive multi-step questions[C]//Proceedings of the 61st annual meeting of the association for computational linguistics (volume 1: long papers). 2023: 10014-10037.

---

> ### Author Response · Authors · 2025-11-20
> **Response to Reviewer TwZX [3/4]**
>
> # Question 1
>
> * We clarify that, because we always expand the path with the smallest retrieval count from a priority queue, the returned path is guaranteed to have minimal retrieval cost under our definition. Here is a detailed explanation for Algorithm 1:
>   * In Line 1, we initialize a priority queue Q, where each element is a trajectory together with its retrieval cost (the number of retrieval actions along that path).
>   * In Line 3, we always remove from Q the trajectory with the lowest retrieval count.
>   * Extending a trajectory by one step can only keep or increase its retrieval count: a retrieval action adds 1 to the cost, and a non-retrieval action adds 0.
>   * Because of this ordering, when we reach Line 7 and find the first trajectory that satisfies IsEqual(o, y), all trajectories with smaller retrieval cost have already been expanded. Any other correct path with a lower retrieval count would have appeared in the queue earlier and would have been removed and checked before . Therefore, the first correct trajectory we return indeed has the minimal retrieval cost among all correct trajectories in this search space.
> * Regarding the reviewer's counterexample :
>   * Path A: [retrieve, retrieve], cost = 2
>   * Path B: [parametric, parametric], cost = 0
>   * Under our implementation, Q is ordered by retrieval count. Thus, Path B (cost 0) will always be taken from the queue and tested before Path A (cost 2). If both paths lead to correct answers, the algorithm will return Path B, not Path A. The situation where Path A is returned first while having a higher cost cannot occur given this priority-queue ordering.
> * We will revise the paper to explicitly state that the queue in Algorithm 1 is ordered by retrieval count and to briefly explain why, under this ordering and our cost definition, the first correct path we return has minimal retrieval cost.
>
>
>
> # Question 2
>
> * Regarding optimal
>   * As clarified in our response to the question 1, Stage I searches over the decision tree and returns a trajectory that has minimal retrieval cost among all correct paths in this tree. In this sense, the trajectories used as supervision in Stage I are indeed optimal with respect to the search objective and search space. We do not claim that Stage I alone already produces a globally optimal parametric policy; instead, it provides high-quality, low-retrieval-cost trajectories as data for subsequent learning.
> * Regarding Stage2 outperform stage 1
>   * Supervised fine-tuning on Stage I trajectories is essentially imitation learning from a fixed set of examples. This has two limitations: 1) It may overfit to the demonstration distribution and may fail to fully generalize to states encountered at test time, where the model’s own generations and retrieval results induce distribution shift. 2) It only teaches the model to copy the expert-like behavior on those specific paths, but does not fully calibrate the model’s knowledge boundary in regions not covered by the Stage I trajectories.
>   * For this reason, we introduce Stage II, where RL-style training further adjusts the policy based on reward signals on the model’s own rollouts. This step explicitly corrects the model on trajectories it actually visits during inference, which improves its ability to decide when internal knowledge is sufficient and when additional retrieval is necessary. As shown in Table 1 (Stage I vs Stage II), Stage II consistently improves over pure SFT on Stage I data, confirming that this extra stage is useful and not redundant. As shown in Table 2, compared to DeepRAG-Imi, DeepRAG-RL further reduces the average number of retrieval calls, confirming that RL provides additional gains in retrieval efficiency beyond imitation learning alone.
> * Regarding RL on discovers better strategies
>   * Within Stage II, we include both RLoff and RLon to provide two types of alignment: RLoff uses data constructed from the Stage I trajectories (off-policy, step-wise dpo ). RLon uses on-policy rollouts sampled from the current model to further refine the policy. Our experiments show that RL (regardless of online or offline) can further correct the model’s internal knowledge and retrieval behavior on top of Stage I. In many cases, on-policy updates provide slightly better calibration of the knowledge boundary, which explains why RLon can sometimes edge out RLoff by a small margin.
>   * We agree that RL can discover better strategies beyond the original Stage I trajectories, because it explores variations around those paths based on on-policy updates. In other words, the rollouts strictly follow the distribution induced by the current policy, which encourages more precise optimization under that distribution. Such improvements do not mean that Stage I trajectories are not optimal with respect to the defined search objective; they simply reflect that the learned parametric policy can be further improved by training on its own rollouts.

---

> ### Author Response · Authors · 2025-11-20
> **Response to Reviewer TwZX [4/4]**
>
> # Question 3
>
> * Sorry for the confusion. We do not use tree search during training. It is only employed for data construction.
> * For the binary tree search, we set the maximum depth to 8 on HotpotQA and 10 on 2WikiMultiHop. If no correct path is found within this limit, the corresponding example is treated as unsolved and removed from Stage I supervision. For DPO, we set β = 0.2, and for GRPO we use a rollout count G = 8. We will open-source all code, hyperparameters, and training scripts after the review process to ensure reproducibility.
>
>
> Thank you once again for your constructive feedback. Although we are unable to directly compare with these baselines, we will include them in our Related Works.
>
> **Dear reviewer TwZX, thank you once again for your feedback. We hope our response has addressed your concerns, and if you have any further questions, please feel free to continue the discussion. We would greatly appreciate it if you could reassess our work and raise your score.**

---

> ### Author Response · Authors · 2025-11-26
>
> Dear Reviewer,
>
> Would you mind taking a look at our responses to check if they address your concerns?
>
> Thank you for dedicating your time and effort to reviewing our work.
>
> Best regards,
>
> Authors of DeepRAG

---

### Meta-Review · Area_Chair_DMXk · 2026-01-05

**Summary:**

The main remaining concern is related to the novelty of this paper (novelty issues by Reviewer taGT and Reviewer JFTj). In particular, this paper’s novelty is incremental compared to related papers but given the impressive performance gain I believe the combination of incremental contributions boost each other.

**Reviewer Concerns:**

**Reviewer TwZX**:
The reviewer shares five concerns and they are addressed as follows:
1. (*presentation*) Overstated Technical Novelty; the "MDP formulation" is superficial packaging and binary search is expansive – properly mitigated concerns by clarifying the reason in using MDP and the efficient heuristic for the binary search.
2. (*narrow experiment scope*) Narrow Evaluation Scope Only tested on artificially constructed multi-hop QA benchmarks – properly addressed by claiming that the evaluation setup follows existing convention for recent related papers.
3. (*missing baseline*) The paper requires three complex stages but never compares to a one-stage baseline – addressed as the existing experiments already have the one-stage baseline.
4. (*algorithm flaw*) An algorithm correctness issue in Line 7 – properly clarified.
5. (*Questionable "Optimality" Claims*) Table 1 shows a counterintuitive result (RLon, which is optimal, is better than RLoff) – clarified by adjusting misunderstanding.

No outstanding remaining concerns.


**Reviewer taGT**:
The reviewer raised four concerns and they are partially addressed as follows:
1. (*novelty*) The method doesn’t seem to offer much innovation – partially addressed by highlighting the binary search step for the data collection of imitation learning.
2. (*fixed retriever*) All experiments assume a fixed Wikipedia retriever – mitigated for claiming fair comparison as known baselines use the fixed Wikipedia retriever.
3. (*verified absence of WebQuestions*) how DeepRAG handles verified absence – explained.
4. (*single-hop performance*) How does the model behave on single-hop factual queries (e.g., PopQA)? – empirically mitigated the concern.

The novelty could be a minor outstanding issue but given the outstanding performance of the paper, this concern could be sufficiently mitigated.

**Reviewer JFTj**:
The reviewer raised four concerns and they are partially addressed as follows:
1. (*novelty*) Determining when to retrieve has already been well studied and the authors do not provide a new solution – partially mitigated by highlighting an MDP retrieval formulation, and the usage of binary tree search for imitation learning, and atomic decision as contributions.
2. (*clarification on experiments 1*) Table 1 shows that Qwen 2.5 32B performs worse than Llama 3 8B, which is confusing for me, could you explain the reason? – addressed by pointing similar trends on all baselines
3. (*clarification on experiments 2*) In the case of Llama, online and offline methods show comparable results. What accounts for this difference？ – explained by introducing similar observations on other papers with interpretation
4. (*clarification on experiments 3*) The majority of queries in HotpotQA and 2Wiki are designed for 2-hop reasoning. Does this imply the presence of significant redundant retrieval? If so, what causes this inefficiency? – explained the misunderstanding by clarifying the difference between 2-hop design for HotpotQA and 2Wiki and the number of retrievals.

The novelty could be a minor outstanding issue but given the outstanding performance of the paper, this concern could be sufficiently mitigated.

**Reviewer Scores:**

**Reviewer TwZX**:
Final expected rating: 8 / final expected confidence: 3 – As concerns are properly addressed, I expect that the review would increase the rating from 6 to 8.

**Reviewer taGT**:
Final expected rating: 6 / final expected confidence: 3 – As most concerns are properly addressed, I expect that the review would increase the rating from 4 to 6.

**Reviewer JFTj**
Final expected rating: 6 / final expected confidence: 4 – As most concerns are properly addressed, I expect that the review would increase the rating from 4 to 6.

---

### Decision · Program_Chairs · 2026-01-26

Accept (Poster)